# Cellular and Molecular Mechanisms in Idiopathic Pulmonary Fibrosis

Yihang Zhang [1,2] and Jiazhen Wang [1,3,*]

1 Collaborative Innovation Center for Chinese Medicine and Respiratory Diseases Co-Constructed by Henan Province and Education Ministry of People's Republic of China, Henan University of Chinese Medicine, Zhengzhou 450046, China
2 The First Clinical Medical College, Henan University of Traditional Chinese Medicine, Zhengzhou 450046, China
3 Academy of Chinese Medicine Science, Henan University of Chinese Medicine, Zhengzhou 450046, China
* Correspondence: jiazhen_wang@hactcm.edu.cn

**Highlights:**

**What are the main findings?**

- We reviewed the latest advances in aberrant molecular events and pathological alterations in different cell populations in idiopathic pulmonary fibrosis.
- We comprehensively summarized the major inducers and signaling pathways of idiopathic pulmonary fibrosis.

**What is the implication of the main finding?**

- It is of great significance to understand the pathological mechanism of idiopathic pulmonary fibrosis.
- To provide new inspiration for the prevention and treatment of idiopathic pulmonary fibrosis.

**Simple Summary:** Idiopathic pulmonary fibrosis is a global disease with unknown etiology. At present, there is still a lack of effective treatment methods, and more in-depth research on this disease is urgently needed. Based on this, we aim to summarize the molecular mechanism and pathological changes of different cell subsets in IPF lung, and review the latest progress of various pro-fibrotic signal transduction pathways in fibrosis, so as to provide key cells and pathways for future research on pulmonary fibrosis, propose more meaningful research directions, and provide theoretical basis for the study of idiopathic pulmonary fibrosis. It is of great significance to understand the pathological mechanism and the prevention and treatment of the disease.

**Abstract:** The respiratory system is a well-organized multicellular organ, and disruption of cellular homeostasis or abnormal tissue repair caused by genetic deficiency and exposure to risk factors lead to life-threatening pulmonary disease including idiopathic pulmonary fibrosis (IPF). Although there is no clear etiology as the name reflected, its pathological progress is closely related to uncoordinated cellular and molecular signals. Here, we review the advances in our understanding of the role of lung tissue cells in IPF pathology including epithelial cells, mesenchymal stem cells, fibroblasts, immune cells, and endothelial cells. These advances summarize the role of various cell components and signaling pathways in the pathogenesis of idiopathic pulmonary fibrosis, which is helpful to further study the pathological mechanism of the disease, provide new opportunities for disease prevention and treatment, and is expected to improve the survival rate and quality of life of patients.

**Keywords:** idiopathic pulmonary fibrosis; alveolar epithelial cells; niche cells; cellular mechanisms; molecular mechanism

## 1. Introduction

Idiopathic pulmonary fibrosis (IPF) is a progressively interstitial lung disease with unknown etiology characterized by interstitial fibrosis, progressive decline of pulmonary

function, dyspnea, and hacking cough [1]. Patients with IPF have a poor prognosis and life quality with a median survival of 2 to 4 years after diagnosis and a higher mortality rate than most patients with malignancies. The increasing morbidity and mortality of IPF impose a severe economic burden on global healthcare [2,3]. The current clinical scheme for IPF mainly includes pharmacologic and nonpharmacologic strategies. The former mainly includes the antifibrotic drugs pirfenidone and nintedanib [4]. Although the two drugs have a certain effect on delaying the decline of lung function, they cannot significantly improve the survival and prognosis of the patients and are prone to developing drug resistance and causing various side effects such as anorexia, vomiting, diarrhea, rash, liver dysfunction, and atherosclerosis. The latter includes oxygen therapy, mechanical ventilation, and lung transplantation. Among them, lung transplantation is currently the most effective treatment for IPF, but only about half of the patients can survive for more than 5 years after receiving lung transplantation. Moreover, due to the lack of donors, the complexity of the surgery, and the high costs, very few patients can really benefit from lung transplantation [5]. Therefore, it is necessary to understand the molecular pathogenesis of IPF and explore potential therapeutic strategies to meet the unmet needs of IPF patients.

The pathological progression of IPF is a dynamic process involving complex interactions among epithelial cells, mesenchymal stem cells (MSCs), fibroblasts, immune cells, and endothelial cells. Single-cell RNA-sequencing analysis from multiple IPF samples revealed that the proportion of airway epithelial cells increased in IPF, while the population of alveolar epithelial cells decreased significantly [6–8]. Activated myofibroblasts and invasive fibroblasts are gradually increased in IPF, and the expression of extracellular matrix (ECM) genes in cells is increased [7,9]. Among the immune cells, the proportion of alveolar macrophages, dendritic cells, and regulatory T cells increased in IPF, and the number of monocyte-like cells and interstitial macrophages decreased [7,10]. Endothelial cells are grouped differently, have different phenotypes, and have different proportions of cells in IPF [11]. Compared with controls, MSCs in IPF exhibited reduced proliferative capacity [12]. Cellular senescence, oxidative stress, endoplasmic reticulum stress, mitochondrial dysfunction, telomere shortening, and aberrant activation of the transforming growth factor-β(TGF-β) pathway are all connected to imbalanced tissue homeostasis (Figure 1).

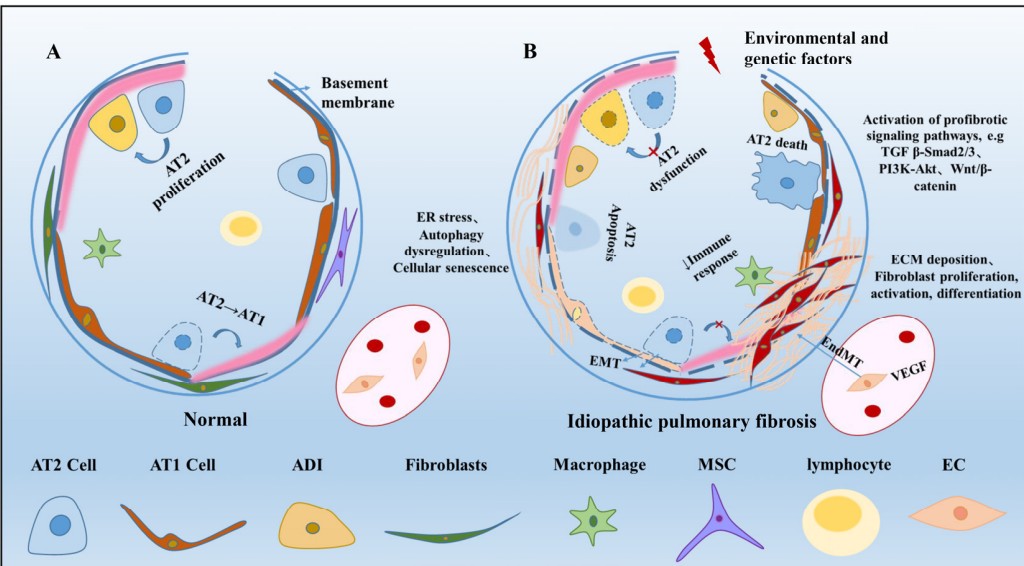

**Figure 1.** Pathological changes in idiopathic pulmonary fibrosis. (**A**) Normally, various cells in the lung work in their own right and work together to maintain lung function and function properly. (**B**) When the lung is exposed to risk factors (e.g., cigarette smoke, genetic or epigenetic alterations, aging)

that cause repeated microdamage to the lung epithelium, inadequate metaplasia of epithelial cells, disruption of the basement membrane, and imbalances in lung homeostasis, the balance between lung and other environmental insults occurs. Aberrant vascular remodeling, epithelial–mesenchymal crosstalk, fibroblast dysplasia, immune attenuation, provocation of profibrotic mediators, activation of fibrotic pathways, deposition of extracellular matrix, and formation of fibrotic foci are induced in this setting. Abbreviations: AT1, alveolar type I epithelial cells; AT2, alveolar type II epithelial cells; ADI, alveolar differentiation intermediate; MSC, mesenchymal stem cells; EC, endothelial cells; ER, endoplasmic reticulum; TGF-β, transforming growth factor-β; ECM, extracellular matrix; EMT, epithelial–mesenchymal transformation; EndMT, endothelial–mesenchymal transition; VEGF, vascular endothelial growth factor.

## 2. Alveolar Epithelial Cells in Pulmonary Fibrosis

An important contributor to the development of idiopathic pulmonary fibrosis (IPF) is the alteration of the intracellular homeostasis of alveolar epithelial cells, which are mainly composed of alveolar type I epithelial cells (AT1), alveolar type II epithelial cells (AT2) [13], as well as abnormal basaloid cells, resulting in aberrant epithelial repair, myofibroblast activation, and increased extracellular matrix deposition to form lung fibrosis [14].

### 2.1. Alveolar Epithelial Type I Cells

AT1 is morphologically flat and covers more than 95% of the surface area of the lung epithelium, which together with the basement membrane, alveolar wall capillaries form the air–blood barrier and are mainly responsible for gas exchange. Efficient gas exchange relies on the intrinsic ion and fluid transport functions of AT1 cells [15]. In addition, AT1 expresses pro-inflammatory receptors, including toll-like receptor 4 (TLR4) and receptors for advanced glycation end-products (RAGE) involved in innate immunity [16,17]. In pulmonary fibrosis, AT1 has received relatively little attention because AT1 is generally considered to be a terminally differentiated cell that does not have the functions of proliferation and differentiation itself [18,19]. However, in contrast to AT2, which is more responsive to hyperoxic injury, AT1 cells rapidly shed after exposure to bleomycin (BLM)-induced lung injury, and are completely lost during the subsequent fibrotic process [20]. There are several lines of evidence supporting that AT1 cells may participate in pulmonary fibrosis. RAGE, which is selectively expressed in AT1 cells and acts as a regulator of inflammation, was found to either promote fibrosis or controversially repress the progression of lung fibrosis [21–24]. Caveolin-1, which encode a scaffold protein of caveolae and is mainly present in AT1, could directly inhibit TGF-β signaling and frequently lost its expression in pulmonary fibrosis [25,26]. Recently, the role of AT1 in alveologenesis and alveolar regeneration is beginning to receive increasing attention. Studies have shown that Hopx+Igfbp2-AT1 cells can be transdifferentiated into AT2 cells to participate in alveolar injury repair [27–30]. These studies have changed the view that AT1 is a terminally differentiated cell. Therefore, the role of AT1 in pulmonary fibrosis needs to be further elucidated (Figure 2).

### 2.2. Alveolar Epithelial Type II Cells

AT2, defined as alveolar stem cells, has the ability of self-renewal and differentiation, which produces pulmonary surfactant, reduces alveolar epithelial surface tension, and prevents alveolar collapse. After alveolar epithelial cell damage, AT2 cells can rapidly proliferate and differentiate into AT1 cells, thereby maintaining the integrity of alveolar epithelial cells, which is necessary for maintaining alveolar homeostasis and promoting alveolar regeneration [31]. Notably, a subset of AT2 cells expressing the transcriptional target of Wnt signaling, Axin2, has been shown to play a leading role in alveolar regeneration and repair, and can be rapidly mobilized, self-renewed, and differentiated into AT1 cells after injury [32]. Disruption of protein homeostasis, telomere damage, mitochondrial dysfunction, and epigenetic changes lead to AT2 dysfunction, manifested as impaired stem cell function, apoptosis, senescence, and pro-fibrotic signaling, which are closely related to the development of IPF and which we will review in detail in this section.

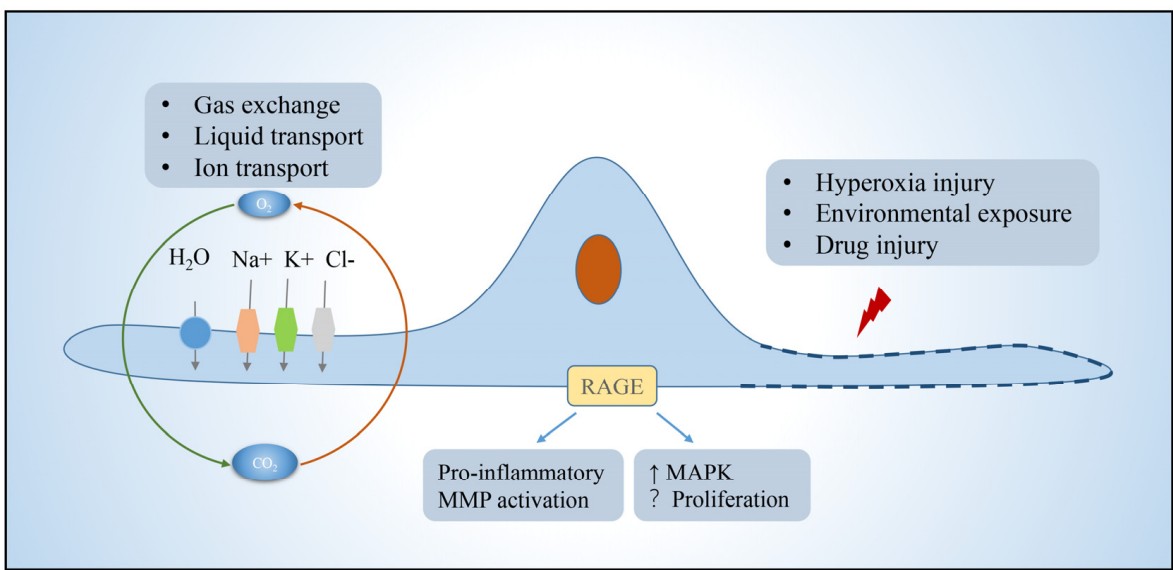

**Figure 2.** Function of alveolar type I epithelial cells. AT1 in the lung is mainly responsible for gas exchange, ion, and fluid transport, and can participate in cellular immunity. AT1 achieves fluid transport via aquaporins, ion transport via the cystic fibrosis transmembrane regulator and ATPases, and proinflammatory, pathway activating, immune, and other functions via a variety of protein receptors such as rage, which is localized to the basement membrane. Function is impaired or cells die when exposed to environmental insults. Abbreviations: RAGE, receptors for advanced glycation end-products; MMP, matrix metalloproteinases; MAPK, mitogen-activated protein kinase.

During the initial injury phase, activated alveolar epithelial cells and recruited inflammatory cells release potent pro-fibrotic growth factors (e.g., TGF-β, tumor necrosis factor α, platelet-derived growth factor PDGF) that induce injury and fibrogenesis [33]. These growth factors, especially TGFβ, are involved in the damage and apoptosis of alveolar epithelial cells, induction of epithelial–mesenchymal transformation (EMT) of alveolar epithelial cells, activation and invasion of fibroblasts, and ECM deposition [34,35]. Human, mouse, and rat alveolar epithelial cells display a pattern of mesenchymal gene expression with the involvement of transcription factors when stimulated with TGFβ in vitro [34]. Sulforaphane (SFN) was found to enhance antioxidant capacity, reverse TGF-β-induced interstitial changes, return cells to epithelioid morphology, and exhibit significant antifibrotic effects on IPF patients, TGF-β-treated cell lines, and BLm-induced fibrosis in mice [36,37]. AT2 cells play a central role in the activation of TGFβ, which may be self-sustained by elevated tension in the fibrotic lung [6]. Cell division cycle 42 (Cdc42) acts on the polymerization of actin in AT2 cells, and its deletion causes a sustained increase in mechanical tension in AT2 cells, leading to activation of the TGF-β pathway, driving the comprehensive development of fibrosis from the periphery to center, with low Cdc42 expression in IPF patient samples [35,38]. Activation of TGF-β in AT2 induces the production of fibrosis, which can be alleviated by inhibition of TGF-β transduction.

### 2.2.1. Protein Homeostasis Destruction

Stemness maintenance of stem cells depends on tightly controlled protein homeostasis [39]. The role of the endoplasmic reticulum (ER) in vivo is to facilitate the folding and transport of proteins to ensure the quality of proteins required for cellular homeostasis. Protein misfolding triggers ER stress, leading to unfolded protein response (UPR), and persistent ER stress causes cellular dysfunction and affects stem cell fate decisions [40,41]. The study found that during pulmonary fibrosis, ER stress marker aggregation and UPR activation existed in AT2 cells. In BLM-injured mice, AT2 cells also exhibited ER stress, and the ER stress activator tunicamycin exacerbates pulmonary fibrosis [42,43]. Deletion of glucose-regulated protein 78 (GRP78), a key regulator of ER homeostasis, causes AT2

cells to undergo ER stress, apoptosis, senescence, impaired stemness, and TGF-β/Smad signaling activation; moreover, GRP78 is severely deleted in AT2 cells of IPF patients [44]. Wang further found that microcystin-leucine arginine (LR) can reduce TGF-β/Smad signaling, regulate macrophage polarization, block EMT and myofibroblast differentiation, and bind to GRP78 to inhibit the UPR signaling pathway, which contributes to alleviating pulmonary fibrosis [45]. ER stress and activation of UPR can cause AT2 dysfunction and further promote the progression of pulmonary fibrosis, which can be alleviated by inhibiting its activation.

### 2.2.2. Mitochondrial Damage

Mitochondria possess their own genetic material and genetic system. In addition to providing energy for cells, mitochondria are also involved in processes such as cell proliferation and differentiation, cell information transmission, and cell apoptosis, and have the ability to regulate cell growth and the cell cycle. Impaired mitochondrial phagocytosis, DNA damage, increased reactive oxygen species, growth disturbance, dysfunction, and disturbance of homeostasis in AT2 cells all induce ER stress and programmed cell death in AT2 cells, leading to the development of fibrosis [46,47]. Mitochondria were damaged and deformed in AT2 cells in IPF compared with control lungs, showing enlargement, swelling, and cristae rupture [48,49]. PTEN-induced putative kinase 1 (Pink1), an enzyme that promotes phagolysis of damaged mitochondria, is downregulated in IPF patients, and the degree of lung fibrosis is increased in Pink1-deficient mice, which is associated with reduced mitophagy, accumulation of malformed mitochondria, ER stress, and increased AT2 apoptotic senescence [43]. Activating transcription factor 3 (Aft3) is highly expressed in fibrotic and aging lungs [48]. Conditional deletion of Aft3 in AT2 protects mice from pulmonary fibrosis [50]. Mitochondrial (mt) DNA base excision repair enzyme, 8-oxoguanine-DNA glycosylase 1 (mtOGG1), can prevent mtDNA damage and apoptosis in epithelial cells. It was found that OGG1-deficient mice had increased pulmonary fibrosis, and IPF patients had increased lung mtDNA damage. mtOGG1 overexpression protected mitochondrial DNA integrity in epithelial cells and weakened cell apoptosis [51]. Phosphoglycerate mutase family member 5 (PGAM5), an important regulator of mitochondrial homeostasis in pulmonary fibrosis, impairs mitochondrial integrity at the functional and structural levels. Ganzleben et al. found that PGAM5-deficient mice and human epithelial cell fibrosis was significantly reduced [52]. These data suggest that improved mitochondrial homeostasis has protective effects on AT2 cells and provides protection against pulmonary fibrosis [53].

### 2.2.3. Telomere Shortening

Telomeres maintain the proliferative potential of stem and progenitor cells by providing a telomerase-dependent repeat expansion mechanism that protects chromosome ends from replicative loss [54]. Telomere shortening impairs stem cell function and tissue regeneration [55]. Telomere shortening in the alveolar epithelium is a common factor in disease progression, and telomere length measured in AT2 is uniformly reduced in IPF [56]. Telomeres are shorter in cells in fibrotic areas compared to non-fibrotic areas in IPF lungs [57]. IPF progression has been shown to be associated with mutations in the telomerase reverse transcriptase family genes, telomerase RNA component (TERC) and telomerase reverse transcriptase (TERT), which regulate telomere length and function [58]. Dysfunctions such as telomere shortening resulting from gene mutations lead to cell cycle arrest, AT2 senescence, and impaired renewal capacity, which are risk factors for the development of IPF [5,59,60]. Recent studies have found that POT1 p. (L259S) is defective in binding telomere protrusion, nuclear accumulation, negative regulation of telomerase, and lagging chain maintenance. Heterozygous mutations in this gene in IPF patients exhibit telomere loss, lagging strand defects, DNA damage, and cellular senescence, and mutations in this gene are thought to be a pathogenic driver of IPF [61]. Overexpression of E3 ubiquitin-protein ligase FBW7 (F-box and WD40 repeat domain-containing 7, also termed FBXW7)

inhibits the expression of telomere capping enzyme tripeptidyl peptidase 1 (TPP1), leading to shortening of telomeres, senescence of AT2 cells, and promotion of the occurrence and development of IPF [62]. Krüppel-like factor 4 (KLF4), a protein transcription factor, is involved in a variety of cellular processes and plays an important role in the maintenance of cellular stemness. The study found that the expression of KLF4 and TERT in IPF patients and fibrosis mouse model AT2 is decreased, while the overexpression of KLF4 can increase the expression of TERT and telomerase activity [63]. Therefore, maintaining the stemness of AT2 stem cells by regulating telomere length and function may be a new way to alleviate the development of fibrosis.

### 2.3. Abnormal Basaloid Cells

In recent years, studies using single-cell sequencing technology and lineage tracing technology have found that there is a previously unknown alveolar differentiation interme­diate (ADI, also called basaloid cells) population in the differentiation progress of AT2 cells to AT1 cells during the repair of alveolar epidermal damage in mice [64,65]. ADI expresses keratin 17/8 (KRT17/KRT8) but does not express the classic AT2 marker Surfactant Protein C (SFTPC) and AT1 markers advanced glycosylation end-product specific receptor (AGER), podoplanin (PDPN), and can further differentiate into mature AT1 cells [64]. In vivo and in vitro functional experiments confirmed that intervention in the ADI-specific regulatory network can promote or inhibit its differentiation into AT1 cells. Surprisingly, the popula­tion of basaloid cells is significantly increased in IPF patients and highly enriched in areas of severe fibrosis [7]. Furthermore, human basaloid cell populations can be transdifferentiated into keratin 5 (KRT5)-positive basal stem cells and their progeny, which largely explains why patients with IPF exhibit an alveolar-bronchialized phenotype [66]. In vitro organoid and in vivo xenograft experiments further confirmed that human basaloid cell populations can differentiate into normal alveolar epithelial cells or transdifferentiate into basal-like cells to promote fibrosis in immunodeficient mice [66] (Figure 3).

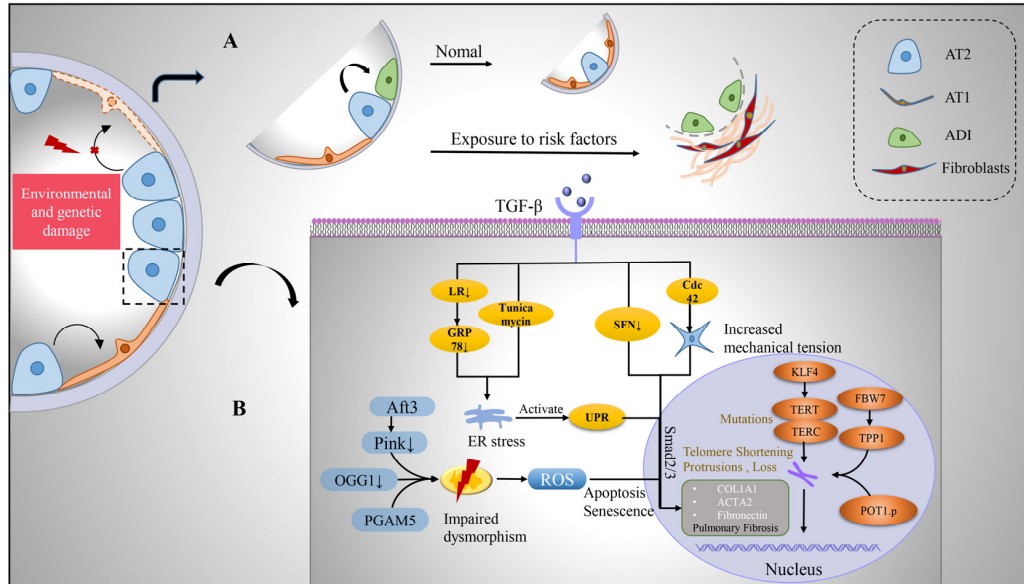

**Figure 3.** Transition of alveolar epithelium and intrinsic pathological mechanism of alveolar type II cells after injury. (**A**) Upon environmental and genetic insults to the alveolar epithelium, AT2 with stem cell functions undergoes self-renewal and differentiation. During its differentiation into AT1, a transitional cell morphology, the ADI, is generated, which is enriched in IPF lungs. (**B**) However, after

AT2 is damaged, pro-fibrosis factor TGF-β is released, which can affect the expression of pro-fibrosis gene, produce endoplasmic reticulum stress, produce UPR, increase cell mechanical pressure, and activate TGFβ-Smad2/3 pathway. Mitochondria in damaged AT2 were damaged and deformed, with enlargement and swelling, ridge breakage, increased reactive oxygen species, inducing apoptosis and senescence of cells. TERT and TERC mutated and telomere shortened and lost under the influence of intracellular genes, which jointly promoted the development of pulmonary fibrosis. Abbreviations: AT1, alveolar type I epithelial cells; AT2, alveolar type II epithelial cells; ADI, alveolar differentiation intermediate; TGF-β, transforming growth factor-β; LR, leucine arginine; ER, endoplasmic reticulum; UPR, unfolded protein response; ROS, reactive oxygen species; GRP78, glucose-regulated protein 78; SFN, sulforaphane; KLF4, Krüppel-like factor 4; TERT, telomerase reverse transcriptase; TERC, telomerase RNA component; FBW7, F-box and WD40 repeat domain-containing 7; TPP1, tripeptidyl peptidase 1; Aft3, Pink1, PTEN-induced putative kinase 1; OGG1, 8-oxoguanine-DNA glycosylase 1; PGAM5, phosphoglycerate mutase family member 5.

## 3. Niche Cells in Pulmonary Fibrosis

Aberrantly activated alveolar epithelial cells may drive the fibrotic response through interactions with niche cells. The proliferation of mesenchymal cells, activation of mesenchymal fibrocytes, recruitment of macrophages and innate lymphoid cells, and mesenchymal transition of endothelial cells can induce the formation of fibroblastic foci that secrete excess extracellular matrix, leading to scarring and destruction of lung architecture.

### 3.1. Mesenchyma Stem Cell

Mesenchymal cells, also known as mesenchymal stem cells, are primitive seed cells with self-renewal and pluripotent differentiation functions [67]. MSCs exist in the perivascular niche in vivo and can participate in processes such as cell development, signal transduction, and cell proliferation in vivo [68]. After local injury, MSCs can be activated and recruited to the injured site, regulate local immune responses, inhibit activated fibroblasts by secreting bioactive molecules (including anti-inflammatory cytokines and chemokines, adhesion molecules, and growth factors), inhibit the apoptosis of epithelial cells and endothelial cells, slow down epithelial–mesenchymal transition, promote epithelial tissue repair, and establish a microenvironment that promotes regeneration [69,70]. Wnt is a key regulator of MSCs and can be induced in IPF, and the sonic hedgehog (Shh) pathway regulates stem cell differentiation. Through experiments, Cao et al. found that blocking the Shh pathway or inhibiting the Wnt protein could prevent the transformation of MSCs into fibroblasts and alleviate pulmonary fibrosis [67].

MSCs can promote tissue repair and immune regulation and inhibit pulmonary fibrosis through various cellular functions [71]. In terms of tissue repair, MSC-derived extracellular vesicles (mEVs) can generate growth factors, proteins, lipids, mitochondria, mRNA, miRNA, DNA, and organelles that can be transferred to damaged recipient cells, thereby promoting re-epithelialization, metabolism, and angiogenesis, contributing to repair [72]. Experiments have found that mEVs can restore adenosine triphosphate (ATP) storage in recipient cells and repair cellular functions by transferring mitochondria from MSCs to damaged cells [73]. In terms of immune regulation, mEVs can play a role by reducing monocyte-induced inflammation, increasing macrophage phagocytosis, reducing neutrophil myeloperoxidase, and inhibiting T-cell proliferation [74]. The cell surface protein Thy-1 is highly expressed in MSCs, mediates mEV expression of miRNAs (such as 199a-3p, 630, 196-5p), and reduces lung fibrosis by inhibiting TGFβ activation and differentiation of lung fibroblasts into myofibroblasts [75]. The levels of miR-199a-5p in IPF patients and BLM-induced pulmonary fibrosis mice were significantly increased. Shi et al. found that miR-199a-5p regulates MSC senescence in IPF patients by regulating the Sirt1/AMPK signaling pathway, which can inhibit MSCs senescence and alleviate fibrosis by downregulating miR-199a-5p expression [76]. MSCs exert immunomodulatory anti-fibrotic effects by reducing the proportion of macrophages with pro-inflammatory and pro-fibrotic cell phenotypes [77,78]. Demonstrated that simultaneous treatment with

MSCs while inducing pulmonary fibrosis in mice with BLM, the model was reversed and pulmonary inflammation as well as pulmonary fibrosis were attenuated [79]. In addition, MSCs directly antagonize the fibrotic process by regulating the ratio of metalloproteinases to tissue inhibitors, thereby reducing the content of collagen fibers, reducing collagen deposition, and inhibiting pulmonary fibrosis [80,81].

Studies have shown that MSCs from IPF patients undergo aging, and the decreased sedimentation capacity leads to DNA damage, mitochondrial dysfunction, and impaired paracrine function, which increases pro-inflammatory responses and disease severity [82,83]. Based on the anti-fibrotic properties of MSCs, studies have shown that MSCs from different sources can inhibit BLM-induced pulmonary fibrosis in mice and promote lung repair through different mechanisms [84]. At the same time, MSC stem cell therapy has been used in clinical trials, and under the condition of guaranteed safety, it has an obvious therapeutic effect on mild to moderate IPF [79,85].

### 3.2. Fibroblasts

Fibroblasts, derived from embryonic mesenchymal cells, are widely distributed, participate in tissue repair, and play a key role in regulating local responses. Activation of lung fibroblasts involves a series of changes in cellular behavior, such as proliferation, migration, and production of ECM, and ECM deposition is a fundamental feature of IPF [86]. The ECM is mainly composed of collagen, elastin, and proteoglycan, which can increase lung compliance, maintain lung tissue structure, and stabilize lung function in the equilibrium state. In an imbalanced state, the composition of the ECM changes, and the amount of fibrillar collagens, proteoglycans, as well as various glycoproteins, especially fibronectin, increases, leading to tissue destruction or fibrosis [87–89]. Boyd et al. found that the production of the ECM protease ADAMTS4 and inflammatory cytokines can induce lung fibroblast damage and immune cell infiltration, which ultimately leads to lung dysfunction and lung microenvironment imbalance [90]. Heat shock protein 90 (Hsp90) regulates multiple processes of fibroblast activation, and its biological activity depends on its ability to bind and hydrolyze ATP. Studies have found that Hsp90 levels and related ATPase activities are elevated in IPF lung and mouse pulmonary fibrosis models, and inhibition of Hsp90 and related ATPase activities can attenuate fibroblast activation, migration, proliferation, invasiveness, attenuate ECM production, and alleviate pulmonary fibrosis, and its inhibitors have been used in the clinical treatment of pulmonary fibrosis [91]. Reducing the production of ECM and its proteases is essential to alleviate fibrosis.

After severe injury, fibroblasts prolifically expand and differentiate into myoblasts. Activation of myoblasts, the main producers of ECMs, leads to excessive deposition of ECMs, scar tissue hyperplasia, destruction of alveolar structures, and irreversible loss of lung function [92]. Potential sources of activated myoblasts include mesenchymal stem cells, epithelial cells, fibroblasts, endothelial cells, and pericytes, in which EMT is the main process of myoblast formation. During EMT, epithelial markers E-cadherin and cytokeratin decreased, while mesenchymal markers N-cadherin, vimentin, α-smooth muscle actin (α-SMA) and fibronectin (FN) increased [93,94]. TGF-β signaling pathway promotes the induction of EMT and the expression of fibrosis-related genes and is involved in the progression of fibrosis. TGF-β mainly relies on the typical Smad signaling pathway: TGF-β induces Smad2/3 phosphorylation to form a complex with Smad4 and translocation to the nucleus to induce the expression of the transcription factor (Snail, Slug, Twist) and trigger EMT. In addition, TGF-β signal transduction can also be used as a non-Smad signal pathway to activate Ras-MAPK, PI3K-Akt, Par6-Smurf1, and other atypical EMT pathways [95,96]. Recent studies have found that abnormal iron metabolism leads to increased cytoplasmic oxidative stress, which can lead to EMT through the activation of autophagy and the expression of pro-fibrosis factors [97,98]. As more and more EMT pathways are being discovered, we believe that inhibition of the TGF-β signaling pathway can inhibit EMT and reduce myofibroblast activation, which may effectively assist the treatment of IPF.

### 3.2.1. Telomere Shortening

Abnormally activated fibroblasts exhibit telomere shortening, metabolic changes, mitochondrial dysfunction, apoptosis resistance, autophagy defects, and senescence-associated secretory phenotype (SASP) secretion, involving multiple molecular signaling pathways [99,100]. Telomere shortening leads to senescence and proliferative arrest of lung fibroblasts. Lung fibroblasts isolated from IPF patients have shorter telomere lengths, and these cells exhibit accelerated replicative senescence during primary culture [101], and telomere shortening is more severe in fibroblasts from patients with genetic defects in telomere homeostasis [102]. Meanwhile, telomerase reverse transcriptase can increase the viability of lung fibroblasts, which is beneficial to the development of fibrosis [103]. Piñeiro's study found that fibroblast activation and collagen deposition were evident, and fibrosis was high in the lungs of telomerase-deficient mice [55].

### 3.2.2. Metabolic Abnormality

Human metabolomic studies have shown that lung tissue from IPF patients exhibits significant differences in energy metabolism, and metabolomic abnormalities in fibroblasts lead to abnormal collagen synthesis and dysregulated airway remodeling [104]. Abnormal glucose metabolism in fibroblasts and alterations in glycolysis contribute to the appearance of features of pulmonary fibrosis [105]. In IPF lungs, upregulated TGF-β signaling in fibroblasts promotes glucose transporter 1 (GLUT1) mRNA expression through the Smad2/3 pathway, activates glycolysis, and the product lactate activates latent TGF-β by altering microenvironmental PH, to meet the energy requirements of abnormal proliferation of fibroblasts and ECM synthesis, and promote fibrosis [106]. Glutamine (Gln) metabolism is required to promote amino acid biosynthesis, and in lung fibroblasts, TGF-β1 upregulates glutaminase expression by activating Smad3 and p38-MAPK-dependent signaling to stimulate glutamine breakdown, maintain cell growth and proliferation, and induce collagen production [107]. At the same time, studies have found that α-ketoglutarate(α-KG), a metabolite of glutaminase, can promote the active expression of jumonji domain-containing protein-3 (JMJD3) in lung fibroblasts and combine with the apoptosis inhibitory proteins X-linked inhibitor of apoptosis (XIAP) and survivin to promote the anti-apoptotic function of IPF fibroblasts [108]. Arginine is involved in collagen synthesis, apoptosis, and ammonia removal, and inhibition of arginase can reduce collagen deposition and improve lung fibrosis [109]. Nitrated fatty acids (NFA), agonists of nuclear hormone receptor peroxisome-activated receptor γ (PPARγ), upregulate PPARγ and block TGFβ-induced fibroblast differentiation, with anti-fibrotic effects [110]. In addition, glycine, arachidonic acid, and succinic acid can all affect IPF through different pathways [111–113].

### 3.2.3. Mitochondrial Damage

Increased mitochondrial reactive oxygen species (ROS) production, mtDNA damage, mitochondrial apoptosis, and senescence in lung fibroblasts can lead to abnormal mitochondrial function and promote fibrosis progression [114,115]. Studies have found that mitochondria in IPF lung fibroblasts have structural abnormalities and an overall decrease in mass and function [101,116,117]. Peroxisome proliferator-activated receptor γ coactivator 1-α (PGC1α) is a transcriptional coactivator that affects mitochondrial biogenesis pathways. PGC1α expression is inhibited in human IPF fibroblasts. Knockout of PGC1α in lung fibroblasts induced pulmonary fibrosis by enhancing fibroblast activation, senescence-related gene expression, and soluble pro- and pro-senescence signals while reducing mitochondrial mass and function [118]. Signal transducer and activator of transcription 3 (STAT3) plays a role in cell cycle progression, gene transcription, and mitochondrial respiration, and is involved in coordinating cellular homeostasis, which is dysregulated during aging [119,120]. Mitochondrial respiration increased after oxidative-induced senescence in fibroblasts, nuclear localization versus mitochondrial localization of STAT3 is observed, and mitochondrial function is restored after STAT3 inhibition, suggesting that STAT3 may serve as a potential molecular target to alter early senescence and restore normal fibroblast

function [121]. In that sense, the structure, quality, and function of mitochondria can greatly affect the function of fibroblasts and the development of IPF.

### 3.2.4. Apoptosis

Apoptosis, the autonomous and orderly death of cells controlled by genes to remove unwanted or abnormal cells, has an important role in maintaining homeostasis and organismal phylogeny. Studies have shown that fibroblasts are the most anti-apoptotic cells in the lung [122], and fibroblasts in IPF lungs are resistant to apoptosis and have reduced apoptosis [101,123]. Cytochrome c, a key signaling molecule during apoptosis, was significantly reduced in IPF fibroblasts. Luis found that this may be related to the reduction of mitochondrial electron transfer, oxygen consumption, and ATP synthesis [116]. A decrease in the pro-apoptotic proteins Bak and Bax and an increase in the anti-apoptotic Bcl-2 family proteins were found in IPF senescent fibroblasts [124]. Scutellarin can regulate the Bcl-2/Bax signaling pathway, inhibit Bcl-2 expression, promote Bax expression, induce fibroblast apoptosis, and alleviate lung fibrosis [125]. Fas is a transmembrane protein that binds to FasL to initiate the transduction of apoptotic signals and induce apoptosis. Caveolin-1 (cav1) is involved in cell signaling in multiple cell types and functions as an anti-fibrotic gene, and Cav-negative fibroblasts are highly resistant to Fas-induced apoptosis and have a greater proportion of α-SMA positivity cells, and XIAP is a key component of fibroblast resistance to Fas-induced apoptosis [122].

### 3.2.5. Autophagy

Autophagy can remove damaged or senescent cellular structures and maintain cellular homeostasis, and senescence in IPF fibroblasts is closely related to autophagy deficiency [126]. Fibroblasts have some classic autophagy pathways involved in the formation of pulmonary fibrosis. PI3K-Akt-mTOR signaling and vimentin intermediate filaments promote lung fibroblast proliferation and pulmonary fibrosis pathogenesis by inhibiting autophagy. Janus kinase 2 (JAK2) and STAT3 inhibit autophagy, leading to abnormal differentiation of fibroblasts and promoting fibrosis progression. Elongation factor-2 kinase (eEF2K) and p38 MAPK activate autophagy and improve abnormal lung fibroblast differentiation, exerting anti-fibrotic effects [127–131]. Autophagy-related biomarkers in lung fibroblasts include apoptotic effector proteins Beclin1, LC3, and p62 [132,133]. As a physiological autophagy inducer, spermidine enhances beclin-1-dependent autophagy and autophagy regulators in IPF fibroblasts and in the lungs of fibrotic mice, significantly reducing inflammation and collagen deposition [134]. Studies have shown that autophagic activity is reduced in the lung tissue of patients with IPF [135,136].

### 3.2.6. Cellular Senescence

Cellular senescence is often accompanied by the generation of the senescence secretory phenotype (SASP), which in IPF senescent fibroblasts includes pro-inflammatory cytokines (e.g., TGF-, IL1β, IL-6, IL-18), chemokines (e.g., CXCL1), growth regulators (e.g., FGF, GM-CSF), matrix metalloproteinases (e.g., MMP-2, MMP-9) [86,100]. IL-18 induces senescence of lung fibroblasts in IPF by blocking the Klotho pathway [137]. Furthermore, the connective tissue growth factor promotes senescence in lung fibroblasts by mediating the accumulation of reactive oxygen species, resulting in the activation of p53 and p16 [138]. Marissa J. et al. evaluated human and murine IPF samples versus control samples and found that p16 expression was consistent with IPF severity and that p16- and SASP-positive fibroblasts aggregated in fibrotic lungs [139]. Recent studies have demonstrated that senescent or IPF fibroblasts can inhibit the proliferation of alveolar epithelial cells, and senescent epithelial cells in the process of IPF can also promote abnormal activation of lung fibroblasts by increasing the expression of SASP [86,140]. The clearance of senescent cells and the blockade of SASP are considered to be new ways of inhibiting IPF (Figure 4).

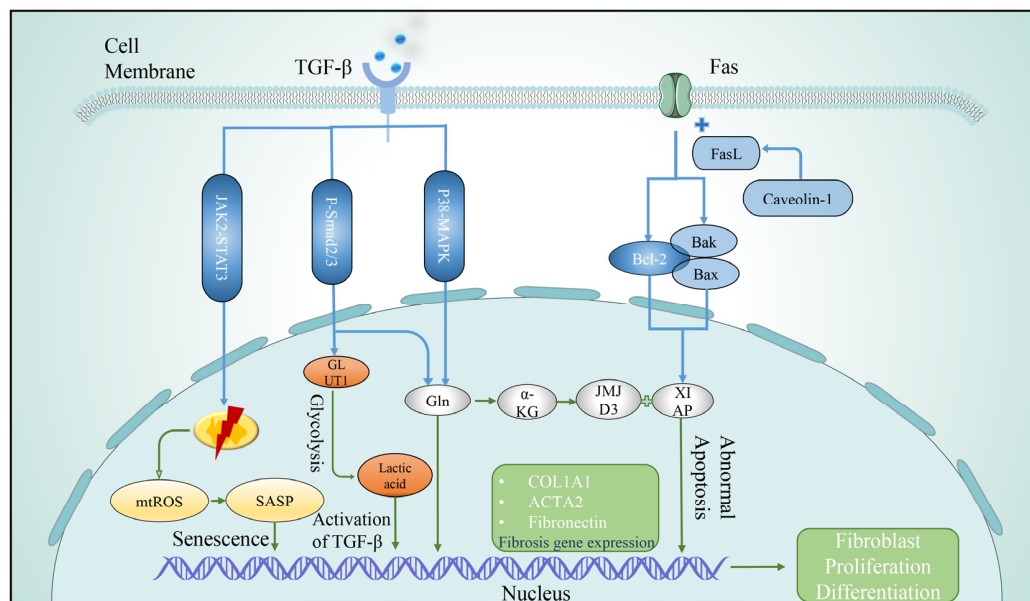

**Figure 4.** Abnormal activation of fibroblasts in idiopathic pulmonary fibrosis. The abnormal activation of fibroblasts in the pathological formation of idiopathic pulmonary fibrosis mainly involves the activation of TGF-β-related pathways JAK2-STAT3, pSmad2/3, and P38-MAPK. Mitochondrial dysfunction leads to increased reactive oxygen species, SASP formation, and cell senescence. Metabonomics such as abnormal metabolism of glucose and amino acids. Fas-FasL apoptosis pathway was inhibited, the expressions of pro-apoptotic proteins Bak and Bax were decreased, and the expressions of anti-apoptotic proteins Bcl-2 and XIAP were increased. The above work together to promote fibroblast proliferation and differentiation and induce fibrosis. Abbreviations: TGF-β, transforming growth factor-β; SASP, senescence-associated secretory phenotype; ROS, reactive oxygen species; GLUT1, glucose transporter 1; Gln, glutamine; α-KG, α-ketoglutarate; JMJD3, jumonji domain-containing protein-3; XIAP, X-linked inhibitor of apoptosis.

### 3.3. Immune Cells

#### 3.3.1. Macrophages

The capacity and specificity of the innate immune response are highly dependent on the regulation and function of multiple cell types. Disruption of macrophage function and excessive apoptosis can lead to abnormal repairs, such as abnormal activation of inflammatory mediators and growth factors, insufficient production of anti-inflammatory macrophages, reduced response to pathogenic stimuli, impaired ability to promote resolution, and impaired communication between macrophages and epithelial, endothelial, fibroblast, and stem cells, all of which contribute to a persistent state of lung tissue damage and promote the development of pathological fibrosis [141,142]. Macrophages can be divided into two types according to anatomical location. One is alveolar macrophages (AMs), which are innate immune cells and key regulatory effectors of tissue repair, regeneration, and fibrosis [142]. AMs play a complex role in pulmonary homeostasis by maintaining the concentrations of alveolar surfactant lipids and proteins. AMs also play an important role in tissue damage control by removing extracellular debris and apoptotic cells and producing anti-inflammatory cytokines such as IL-10 [143,144]. The phagocytic capacity of AMs declines with age, leading to impaired clearance of pathogens in the lungs [145,146]. New studies have shown that histone deacetylase 3 (HDAC3) is a key epigenetic factor for AMs maturation and homeostasis, and HDAC3-deficient AMs exhibit severe mitochondrial oxidative dysfunction and cell death [147]. The other is interstitial macrophages (IMs), which play an important role in immune regulation [148]. IMs are involved in the abnormal activation of fibroblasts that produce ECM, an important driver of fibrosis. Macrophages are important producers of matrix metalloproteinases (MMPs), which can degrade ECM, but some of them, such as MMP12, have bidirectional regulation on ECM formation. IL-13

can promote fibrosis by increasing macrophage metalloelastase activity [149,150]. The activating protein-1 (AP-1) protein can activate or repress gene transcription and participate in the development of various chronic diseases by regulating cellular processes. Ucero et al. found that the AP-1 transcription factor Fra-2 can specifically control the paracrine profibrotic activity of macrophages through the transcriptional target ColVI. Fra-2 transgenic (Fra-2Tg) mice exhibit spontaneous pulmonary fibrosis, and administration of Fra-2 inhibitors reduces ColVI expression and improves fibrosis in Fra-2Tg mice and a mouse model of pulmonary fibrosis [151]. Recent studies have shown that SARS-CoV-2 can trigger a pro-fibrotic transcriptional phenotype in macrophages and induce pulmonary fibrosis [152].

### 3.3.2. Lymphocytes

Lymphocytes are an important cellular component involved in the immune response function of the body and play an important role in resisting external infections and monitoring cell variation in the body. According to their migration, surface molecules, and functions, they can be divided into innate lymphocytes, T lymphocytes (T cells), B lymphocytes (B cells), and natural killer (NK) cells. Innate lymphocytes (ILCs) interact with epithelial cells, T cells, and myeloid cells in the lung to form an immune system network [153]. ILCs are divided into three subgroups, in which ILC2s play a key role in lung homeostasis, lung repair, inflammation, and immune response processes, and their cellular dysfunction affects the progression of IPF [154]. Regnase-1 is considered to be a key transcriptional regulator of the pro-fibrotic function of ILC2s and can inhibit the mRNA expression of Gata3 and Egr1, the transcription factors that regulate fibrosis genes. Regnase-1 deficiency can lead to spontaneous proliferation and activation of ILC2s in the lung, enhancing the degree of fibrosis [155]. IL-5 produced by ILC2s can also play a role in preventing lung injury or mediating repair. Hrusch et al. found that the expression of the inducible T-cell costimulatory molecule (ICOS), which is important for maintaining lung barrier function, was low and reduced ILC2s in IPF patients and the BLM-induced murine fibrosis model, while mouse mortality was reduced after treatment with IL-5 [156]. These data indicate that the immune regulatory function of ILC2s plays a role in relieving the development of fibrosis.

### 3.3.3. T Cells

Among T cells, the role of Th1, Th2, and Th17 is the most studied in relation to pulmonary fibrosis. Th1 cells and their secreted products are considered anti-fibrotic, while Th2 responses lead to tissue damage and produce pro-fibrotic effects [157]. Th2 cell-derived cytokines, including IL-31, contribute to inflammatory and fibrotic remodeling in lung tissue. The expression of IL-31 is elevated in human IPF lungs, and blockade of IL-31 signaling inhibits collagen deposition, attenuates the decline in lung function, and improves pulmonary fibrosis [158]. Th17 cells produce cytokines, such as IL-17, that stimulate ECM production, collagen deposition, regulate TGF-β signaling, and induce pulmonary fibrosis [159]. Studies have found that Th17 is the largest T-cell subset expressing programmed cell death 1 (PD-1), and PD-1+ Th17 cells exist in pulmonary fibrosis of different etiologies. PD-1 exerts profibrotic effects by enhancing STAT3 expression to produce profibrotic cytokines, such as TGF-β and IL-17A [159,160]. T-cell immunoglobulin domain and mucin domain-3 (TIM-3) is mainly expressed on the surface of activated Th1, Th17, and macrophages, and is a novel immunomodulatory protein of the TIM family. Experiments found that the expression of TIM-3 was significantly increased in fibrotic lungs. Overexpression of TIM-3 can induce macrophages to secrete more TGF-β1 and IL-10, aggravating the pathological changes in pulmonary fibrosis, and regulating the expression of TIM-3 can regulate the development of pulmonary fibrosis [161]. The influence of different subsets of T cells on IPF needs more research and exploration.

### 3.3.4. B Cells

The B cells represent another branch of the adaptive immune system. Multiple studies have demonstrated increased B cell activation in IPF lungs [162–164]. Activation of B cells through pattern recognition receptors (PRRs) promotes the release of inflammatory cytokines, chemokines, and metalloproteinases, contributing to the development of pulmonary fibrosis. CpG and β-glucan stimulation of B cells through PRRs leads to activation of the mTOR pathway, increased fibroblast migration and aberrant activation, and promotes inflammatory and fibrotic changes in IPF patients [165].

### 3.4. Endothelial Cells

Endothelial cells (ECs) are one of the most important cellular components of blood vessels and can synthesize and secrete mediators, such as chemokines and lipid mediators, which play a key role in angiogenesis, regulating immune responses, maintaining tissue integrity, barrier function, and preserving cellular communication [166]. Abnormal angiogenesis is a significant pathological feature of IPF. Vascular endothelial growth factor (VEGF) can play a role in angiogenesis, anti-inflammatory, alveolar epithelial growth, and proliferation, preventing epithelial and endothelial cell apoptosis to promote the repair after lung injury and inhibit pulmonary fibrosis [167]. Studies have found that VEGF expression is reduced in IPF patients and mouse models, and VEGF overexpression can reduce mortality in mice with pulmonary fibrosis [168]. In addition, different subtypes of VEGF have different regulatory effects on IPF, such as increased expression of VEGF-A165b in IPF, and anti-VEGF antibody CBO-P11 can significantly reduce BLM-induced pulmonary fibrosis [169].

ECs were divided into multiple subsets. By single-cell RNA sequencing analysis, the subpopulation with high C-X-C motif chemokine ligand 12 (Cxcl12) expression and low nitric oxide synthase 3 (Nos3) expression showed a pro-fibrotic phenotype that was enriched in biological processes related to lung injury and fibrosis, suggesting its role in recruiting monocytes, inducing fibroblast proliferation, and promoting the ECM, and may play a key role in pulmonary fibrosis through potential cross-linking with AMs and stromal cells [11]. In contrast, the subgroup with high Nos3 expression and low Cxcl12 expression was denser in the control lung, expressing its role in regulating physiological functions such as cell adhesion, proliferation, and the injury repair process [170].

Endothelial-mesenchymal transition (EndMT) is an important process of fibrosis development in IPF. During EndMT, ECs lose the expression of vascular endothelial cadherin and other specific markers, obtain fibroblast-like mesenchymal phenotype to express α-SMA, vimentin, and type I collagen, which are further transformed into fibroblasts and promote the development of fibrosis [92,94]. Jia et al. found that ECs decreased in BLM-induced lung fibrosis, while fibroblast marker proteins increased. scRNA-seq revealed that genes complement C3a receptor 1 (C3ar1) and galectin-3, which play a key role in EC dynamic transition, were ubiquitously expressed by ECs in BLM-induced lung fibrosis, and ECM deposition was reduced and lung fibrosis was relieved after inhibition of this gene expression [171]. Sterol regulatory element-binding protein 2 (SREBP2) can play a role in IPF by modifying the EC phenotype. The expression of SREBP2 increased in lung samples of IPF patients, and SREBP2 overexpression induces EndMT, leading to activation of TGF and Wnt signaling, increased ECM deposition, and aggravation of pulmonary fibrosis [172] (Figure 5). By this token, it seems that inhibiting the transformation of EC into fibroblasts by hindering the EndMT process can slow the development of IPF.

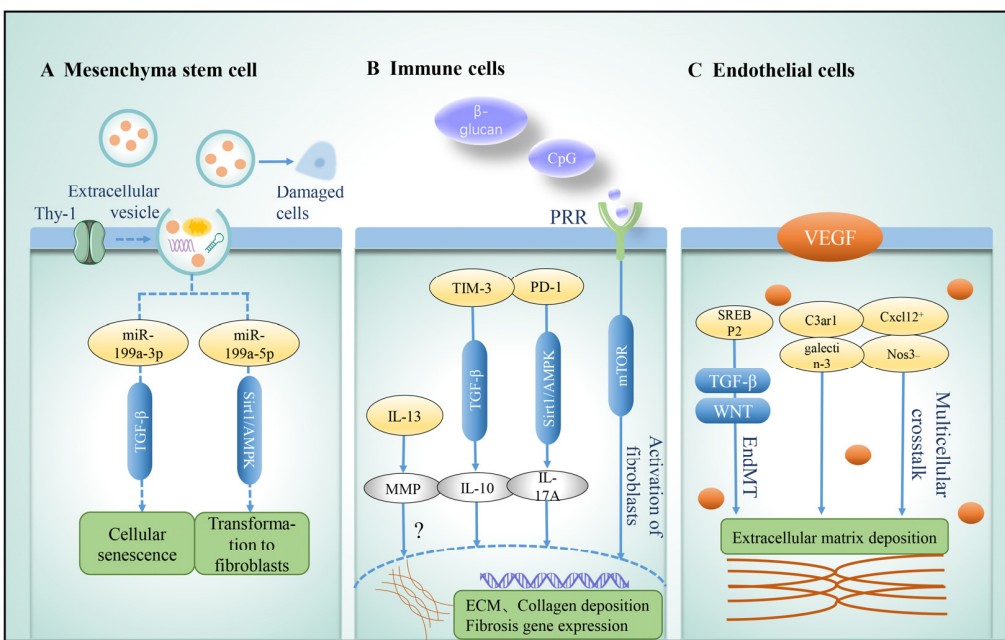

**Figure 5.** Role of niche cells in idiopathic pulmonary fibrosis. Niche cells play different roles in the occurrence and development of idiopathic pulmonary fibrosis. (**A**) MSC-derived extracellular vesicles (mEV) produce a variety of growth factors and genetic material, transfer to damaged cells to repair cellular functions, and regulate intracellular signaling pathways (TGF-β, Sirt1/AMPK) to play an immunomodulatory role. (**B**) Immune cells have bidirectional regulatory effects, and their derived cytokines (IL-13, IL-10, and IL-17A) can not only promote the development of fibrosis through the activation of TGF-β, Sirt1/AMPK, mTOR, and other pathways but also play a role in immune regulation, tissue repair, and degradation of ECM. (**C**) The role of endothelial cells is mainly reflected in barrier function and cell communication. VEGF promotes lung repair after injury, but its different subtypes have different regulatory effects on IPF. For example, subgroups with high expression of Cxcl12 and low expression of Nos3 show pro-fibrosis phenotype. Endothelial–mesenchymal transition is an important process in the development of fibrosis. In this process, C3ar1 and galectin-3 are the key genes, TGF and Wnt signals are activated, ECM deposition is increased, and pulmonary fibrosis is aggravated. Abbreviations: TGF-β, transforming growth factor-β; ECM, extracellular matrix; AMPK, adenosine 5′-monophosphate (AMP)-activated protein kinase; VEGF, vascular endothelial growth factor; EndMT, endothelial–mesenchymal transition; PRR, pattern recognition receptors; MMP, matrix metalloproteinases; TIM-3, T-cell immunoglobulin domain and mucin domain-3; PD-1, programmed cell death 1; SREBP2, sterol regulatory element-binding protein 2; C3ar1, C3a receptor 1; Cxcl12, C-X-C motif chemokine ligand 12; Nos3, nitric oxide synthase 3.

## 4. Cellular Crosstalk

In conclusion, in the process of IPF development, different cells have their own roles (Table 1), and intercellular crosstalk is ubiquitous. As one of the cells that play a major role in IPF, once the alveolar epithelium is damaged, AT2 proliferates and differentiates into AT1, which can secrete a variety of cytokines and chemokines while repairing the damage, stimulate the recruitment and activation of immune cells at the injured site, and promote the abnormal activation of lung fibroblasts and the transformation of myofibroblasts [45,173,174]. Fibroblasts are mainly transformed from epithelial cells, mesenchymal cells, and endothelial cells through EMT and EndMT pathways [94]. Dysfunctional fibroblasts, in turn, inhibit the proliferation and differentiation of alveolar epithelial cells, induce inflammation in the cellular microenvironment through the secretion of pro-inflammatory cytokines, promote the accumulation of immune cells, and impede tissue repair [175,176]. The dysfunction of immune cell function and excessive cell apoptosis can lead to abnormal repair and damage of the immune network, resulting in communication barriers between

epithelial cells, endothelial cells, fibroblasts, and stem cells, leading to persistent injury of lung tissue and promoting the development of pathological fibrosis [141,165]. ECs transform into mesenchymal cells and further into fibroblasts during EndMT. Endothelial cell dysfunction leads to irreversible vascular remodeling and increased vascular resistance in fibrosis, leading to the development of pulmonary hypertension, promoting the transformation into mesenchymal cells, and increasing fibrogenesis. Its fibrogenic subgroup can recruit immune cells, crosslink stromal cells, induce fibroblast proliferation, promote ECM deposition, and promote the development of fibrosis [11,177]. MSC can inhibit activated fibroblasts, inhibit apoptosis of epithelial cells and endothelial cells, slow down epithelial–mesenchymal transformation, regulate immune cells to play their role in immune regulation, promote tissue repair, promote microenvironment regeneration, and alleviate fibrosis through derived extracellular vesicles and secretion of cytokines [69,178].

**Table 1.** Effects of various cells in IPF.

| Cells | Function and Mechanism | Effects in Lung Disease | Ref. |
|---|---|---|---|
| AT1 | Gas exchange, Ion and liquid transport, Congenital immunity | Compositional gas barrier, Involved in inflammation | [15–17] |
| AT2 | Self-renewal and differentiation, Damage repair | Impaired stem cell function, Pro-fibrotic signaling, ER stress, Telomere attrition, Mitochondrial dysfunction, Differentiate into fibroblasts | [31,33,39–41,46,47,54–56] |
| Abnormal basaloid cells | Transdifferentiation | Alveolar bronchogenic phenotype | [64,65] |
| MSC | Self-renewal and differentiation, Damage repair, Immunoregulation | Secreted bioactive molecules, Inhibited fibroblast activation, Inhibited cell apoptosis, Reduced epithelial-mesenchymal transition | [67–71] |
| Fibroblasts | Tissue repair | ECM deposition, Differentiate into myoblasts, Telomere shortening, Metabolic abnormality, Mitochondrial damage, Resistance to apoptosis, Autophagy, Cellular senescence | [86–89,92–96,99,100,104,114, 115,122,123,126] |
| Immune Cells | Damage repair, Immunoregulation | Promote the release of inflammatory factors, ECM deposition/Degradation of ECM | [141,142,153,154,157,162–164] |
| ECs | Angiogenesis, Damage repair, Immunoregulation | Anti-inflammation, Inhibited cell apoptosis/Differentiate into fibroblasts | [92,94,166,167] |

Abbreviations: AT1, alveolar type I epithelial cells; AT2, alveolar type II epithelial cells; MSC, mesenchymal stem cells; ECs, endothelial cells; ER, endoplasmic reticulum; ECM, extracellular matrix.

## 5. Discussion and Perspectives

Idiopathic pulmonary fibrosis is a global disease with an unknown etiology that has been widely explored. In recent years, our insights into the pathogenesis of IPF have been profoundly revolutionized by the use of emerging technologies such as single-cell sequencing. This review summarized the molecular mechanism and pathological changes in different cell subsets in the IPF lung, highlighting that aging, apoptosis resistance, autophagy defects, organelle dysfunction, metabolic abnormalities, and epigenetic changes are the main inducing factors of IPF, whereas the TGF-β/Smads pathway, Fas/FasL apoptotic pathway, and PI3K/Akt signal transduction pathway also play important roles in the development of IPF. This complex network is jointly involved in the progression of IPF disease, which is of great significance for understanding the pathological mechanism of the disease and the prevention and treatment of the disease. In the future, we will focus on the regulation mechanism of TGF-β-related pathways and the effect of intercellular

communication on fibrosis, especially the effect of niche cells on the maintenance of alveolar stem cell function. In the future, the challenge will be to regulate cell morphology and function with pro-fibrotic signaling pathways in vivo, combined with targeted cells and key pathway therapeutic drugs, hoping to bring safe, novel, and effective treatment opportunities for idiopathic pulmonary fibrosis.

**Author Contributions:** Conceptualization, J.W. and writing—original draft preparation, Y.Z.; writing—review and editing, Y.Z.; visualization, Y.Z.; supervision, J.W.; project administration, J.W.; funding acquisition, J.W. All authors have read and agreed to the published version of the manuscript.

**Funding:** This research was funded by the China National Postdoctoral Program for Innovative Talents, grant number BX2021093.

**Institutional Review Board Statement:** Not applicable.

**Informed Consent Statement:** Not applicable.

**Data Availability Statement:** Not applicable.

**Acknowledgments:** I am grateful to my colleagues Yanmei Yang for assistance with some of the studies described and Fan Yang for intellectual input to the direction of graph edit. Most importantly, the work I have outlined involved major contributions from the following current members of my laboratory (listed alphabetically): Jiahui Huang, Jiayi Wang, Ru Wang.

**Conflicts of Interest:** The authors declare no conflict of interest, financial or otherwise, are declared by the author.

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
