# Peer review of "Cellular and Molecular Mechanisms in Idiopathic Pulmonary Fibrosis"

_arm, doi:10.3390/arm91010005_

Round 1
Reviewer 1 Report
Yihang Zhang and Jiazhen Wang have tried to understand cellular and molecular mechanisms in idiopathic pulmonary fibrosis. The review is written well and is suitable for publication.
1. What is the main question addressed by the research?
The research addresses the latest knowledge on how molecular and pathological changes in different cell populations is associated with idiopathic pulmonary fibrosis
.
2. Do you consider the topic original or relevant in the field? Does it
address a specific gap in the field?
Yes, the topic is relevant in the field and addresses the gap by adding new advances on the role of different cells in IPF.
3. What does it add to the subject area compared with other published material?
It added information from published papers of the recent years on IPF.
4. What specific improvements should the authors consider regarding the
methodology? What further controls should be considered?
NA
5. Are the conclusions consistent with the evidence and arguments presented
and do they address the main question posed?
Yes
6. Are the references appropriate?
Yes
7. Please include any additional comments on the tables and figures.
NA
Author Response
We thank the reviewer for the time and effort in reviewing the previous version of the manuscript, and we are also very grateful to the reviewers for the affirmation of our work. We will carefully revise the manuscript according to comments, so as to further meet the requirements for successful publication. Thank the reviewer again for the comments.
Comment 1: What is the main question addressed by the research?
The research addresses the latest knowledge on how molecular and pathological changes in different cell populations is associated with idiopathic pulmonary fibrosis.
Response 1: Thanks for the reviewer's positive feedback.
Comment 2: Do you consider the topic original or relevant in the field? Does it address a specific gap in the field?
Yes, the topic is relevant in the field and addresses the gap by adding new advances on the role of different cells in IPF.
Response 2: Thanks for the reviewer's positive feedback. I'm glad to get such a comment.
Comment 3: What does it add to the subject area compared with other published material?
It added information from published papers of the recent years on IPF.
Response 3: Thanks for the reviewer's positive feedback. I hope this review will make a contribution to the field.
Comment 4: What specific improvements should the authors consider regarding the methodology? What further controls should be considered?
NA
Response 4: Thanks for the reviewer's positive feedback. we will be carefully revised according to the comments of all reviewers
Comment 5: Are the conclusions consistent with the evidence and arguments presented and do they address the main question posed?
Yes
Response 5: Thanks for the reviewer's positive feedback. I'm glad to get such a comment.
Comment 6: Are the references appropriate?
Yes
Response 6: Thanks for the reviewer's positive feedback. If there are unreasonable references, they have been revised in time.
Comment 7: Please include any additional comments on the tables and figures.
NA
Response 7: Thanks for the reviewer's positive feedback.
Reviewer 2 Report
The paper submitted for review discusses very well the different cell types and molecular processes involved in the development of IPF. Current experimental, review and clinical trials literature as well as case reports were used, which is very interesting as it integrates basic research with clinical trials, which has a key impact on the development of science. However, I have a few comments, which I outline below:
1) It seems to me that a good review paper should refer mainly to experimental papers (preferably the most recent but not only), in the paper submitted for review almost 30% (29.6%) are review papers (48/162: 2, 3, 5-7, 12, 13, 25, 27, 30, 32-35, 40-44, 49, 52, 53, 56, 57, 67, 68, 71, 72, 75, 88, 89, 92, 96, 97, 106, 107, 117, 118, 120, 131, 132, 137, 142, 143, 146, 156, 159, 160) to which the authors refer. Please review all these papers carefully and change the review papers to experimental papers accordingly, so as to reduce the number of review papers as much as possible (1-2 papers per subsection on a particular cell type is sufficient).
2) Please correct the typos in Fig 1 and enlarge the inscriptions accordingly, as they are not very legible (also in other figures)
3) No references in the text to Fig 5A, 5B, 5C - please correct
4) Please highlight the descriptions to figures 1-5 and write any abbreviations used on the figures in the descriptions.
5) Please read the text carefully and explain all necessary abbreviations (also those that appear in the figures)
6) I miss a diagram/table summarising the content presented - integrating all the described cellular and molecular mechanisms influencing the development of IPF - it should be simple, but illustrate the interactions of the different cell types (and the processes taking place in them very well described in the text)
7) It seems strange to me to mention COVID-19 pandemic in the abstract and to completely fail to address this thread in the text of the paper - please make a few words of comment (expand the discussion to include this aspect in the context of the work and results discussed) or remove this section from the abstract.
8) In the subsection on mesenchymal cells (MSCs and fibroblasts), I very much miss a brief discussion of the phenotypic transition of fibroblasts to myofibroblasts (and the EMT process), which are crucial for the development of IPF as well as other diseases where fibrotic processes play a key role. Please provide a few sentences describing this process together with a listing of myofibroblast markers (literature can be used e.g. Michalik, M., Wójcik-PszczoÅ‚a, K., Paw, M., Wnuk, D., Koczurkiewicz, P., Sanak, M., PÄ™kala, E., & Madeja, Z. (2018). Fibroblast-to-myofibroblast transition in bronchial asthma. Cellular and molecular life sciences : CMLS, 75(21), 3943-3961. https://doi.org/10.1007/s00018-018-2899-4) and a brief characterisation of these cells in IPF.
9) In my opinion, the topic shown in the paper is well discussed and the authors should focus on the interactions between the different cell types, while in the introduction of each subsection on a particular cell type, point to specific review/experimental work on the topic, e.g.:
- epithelial cells: Chakraborty, A., Mastalerz, M., Ansari, M., Schiller, H. B., & Staab-Weijnitz, C. A. (2022). Emerging Roles of Airway Epithelial Cells in Idiopathic Pulmonary Fibrosis. Cells, 11(6), 1050. https://doi.org/10.3390/cells11061050
- imune cells: Desai, O., et al, The Role of Immune and Inflammatory Cells in Idiopathic Pulmonary Fibrosis. Front Med (Lausanne), 820 2018. 5: p. 43.
- Fibroblasts: Ramos, C., Montaño, M., García-Alvarez, J., Ruiz, V., Uhal, B. D., Selman, M., & Pardo, A. (2001). Fibroblasts from idiopathic pulmonary fibrosis and normal lungs differ in growth rate, apoptosis, and tissue inhibitor of metalloproteinases expression. American journal of respiratory cell and molecular biology, 24(5), 591-598. https://doi.org/10.1165/ajrcmb.24.5.4333
Lin, Y., & Xu, Z. (2020). Fibroblast Senescence in Idiopathic Pulmonary Fibrosis. Frontiers in cell and developmental biology, 8, 593283. https://doi.org/10.3389/fcell.2020.593283
ect.
I believe that with a few minor revisions it will be possible to publish the article: 'Cellular and molecular mechanisms in Idiopathic pulmonary fibrosis' in the journal: Advances in Respiratory Medicine.
Author Response
We thank the reviewer for the positive comments and valuable suggestions to increase the potential impact of our work. We carefully considered all the comments and further revised and explained them to fully solve the problems raised by the reviewer in our revised version.
Comment 1:It seems to me that a good review paper should refer mainly to experimental papers (preferably the most recent but not only), in the paper submitted for review almost 30% (29.6%) are review papers (48/162: 2, 3, 5-7, 12, 13, 25, 27, 30, 32-35, 40-44, 49, 52, 53, 56, 57, 67, 68, 71, 72, 75, 88, 89, 92, 96, 97, 106, 107, 117, 118, 120, 131, 132, 137, 142, 143, 146, 156, 159, 160) to which the authors refer. Please review all these papers carefully and change the review papers to experimental papers accordingly, so as to reduce the number of review papers as much as possible (1-2 papers per subsection on a particular cell type is sufficient).
Response 1: Thank you very much for the valuable advice. The original intention of citing review references is to provide more background knowledge. We followed the advice and inappropriate review references have been modified. The review papers account for 13.5%(24/178:1,5,13-15,39,46,47,54,59,60,92,99,103,104,113,114,141,152,156,165,173,175).
Comment 2:Please correct the typos in Fig 1 and enlarge the inscriptions accordingly, as they are not very legible (also in other figures)
Response 2: I am very sorry for the mistake and the trouble caused to your review. The mistake has been corrected and the font in the picture has been enlarged. Thank you for your advice.
Comment 3:No references in the text to Fig 5A, 5B, 5C - please correct
Response 3: Thank you for your question. The reference corresponding to Figure 5A is 70-79, the reference corresponding to Figure 5B is 140-163, and the reference corresponding to Figure 5C is 165-170, Thank you for your review.
Comment 4:Please highlight the descriptions to figures 1-5 and write any abbreviations used on the figures in the descriptions.
Response 4: I'm very sorry for such negligence and omission. The font has been enlarged or boldened, and the abbreviations have been added. Thank you for your advice.
Comment 5:Please read the text carefully and explain all necessary abbreviations (also those that appear in the figures)
Response 5: I'm sorry for such omission. I have carefully read the full text and explained abbreviations. Thank you for your advice.
Comment 6:I miss a diagram/table summarising the content presented - integrating all the described cellular and molecular mechanisms influencing the development of IPF - it should be simple, but illustrate the interactions of the different cell types (and the processes taking place in them very well described in the text)
Response 6:Thanks for the reviewer's positive comment, I have added a table as a summary, and the article is clearer. Thank you for your valuable suggestion.
Comment 7:It seems strange to me to mention COVID-19 pandemic in the abstract and to completely fail to address this thread in the text of the paper - please make a few words of comment (expand the discussion to include this aspect in the context of the work and results discussed) or remove this section from the abstract.
Response 7: Thank you for your valuable comment. The content of COVID-19 pandemic is really not suitable for the abstract. It has been deleted. Thank you for your reminding.
Comment 8:In the subsection on mesenchymal cells (MSCs and fibroblasts), I very much miss a brief discussion of the phenotypic transition of fibroblasts to myofibroblasts (and the EMT process), which are crucial for the development of IPF as well as other diseases where fibrotic processes play a key role. Please provide a few sentences describing this process together with a listing of myofibroblast markers (literature can be used e.g. Michalik, M., Wójcik-PszczoÅ‚a, K., Paw, M., Wnuk, D., Koczurkiewicz, P., Sanak, M., PÄ™kala, E., & Madeja, Z. (2018). Fibroblast-to-myofibroblast transition in bronchial asthma. Cellular and molecular life sciences : CMLS, 75(21), 3943-3961. https://doi.org/10.1007/s00018-018-2899-4) and a brief characterisation of these cells in IPF.
Response 8: Thanks to the reviewer for the valuable comment, which enriched and complete the content of my paper. I also appreciate the reference provided by the reviewer, from which I learned a lot. I have added the content of fibroblast transformation into myofibroblast and EMT. Thank you for your help.
Comment 9:In my opinion, the topic shown in the paper is well discussed and the authors should focus on the interactions between the different cell types, while in the introduction of each subsection on a particular cell type, point to specific review/experimental work on the topic, e.g.:
- epithelial cells: Chakraborty, A., Mastalerz, M., Ansari, M., Schiller, H. B., & Staab-Weijnitz, C. A. (2022). Emerging Roles of Airway Epithelial Cells in Idiopathic Pulmonary Fibrosis. Cells, 11(6), 1050. https://doi.org/10.3390/cells11061050
- imune cells: Desai, O., et al, The Role of Immune and Inflammatory Cells in Idiopathic Pulmonary Fibrosis. Front Med (Lausanne), 820 2018. 5: p. 43.
- Fibroblasts: Ramos, C., Montaño, M., García-Alvarez, J., Ruiz, V., Uhal, B. D., Selman, M., & Pardo, A. (2001). Fibroblasts from idiopathic pulmonary fibrosis and normal lungs differ in growth rate, apoptosis, and tissue inhibitor of metalloproteinases expression. American journal of respiratory cell and molecular biology, 24(5), 591-598. https://doi.org/10.1165/ajrcmb.24.5.4333
-Lin, Y., & Xu, Z. (2020). Fibroblast Senescence in Idiopathic Pulmonary Fibrosis. Frontiers in cell and developmental biology, 8, 593283. https://doi.org/10.3389/fcell.2020.593283 ect.
Response 9: Thank the reviewers for the valuable suggestions and references, which have greatly helped to enrich my article. Each part of the paper contains a description of the intercellular interactions. In the last part of the paper, I summarized the intercellular crosstalk as suggested. Thank you for your advice.
Reviewer 3 Report
Zhang and wang have presented a literature review that summarises the latest advances in idiopathic pulmonary fibrosis (IPF). They discuss the different cell types and explore the pathological changes and aberrant events that contribute to the development of IPF. As pointed out It is of significant interest to understand the molecular and pathological changes that leads to development of IPF to potentially developed new drugs to target IPF. While this review is interesting major revision is necessary before it can be considered for publication.
Abstract (lines 32-33)
Need to remove the reference to COVID-19 as there is only 1 mentioning of COVID-19 in the manuscript itself and that one sentence is out of place as well. Reword the last sentence as well to better reflect the conclusion and/or take-home message for the readers.
Introduction:
First paragraph is written quite well with a good flow of information and why you write the review. The second and third paragraph is quite dense with information. This needs some rewording to improve the flow and improve readability for the reader.
On page 3 line 75 what do the authors mean with “work in their own right”?
Alveolar epithelial cells in pulmonary fibrosis:
What do the authors mean in the context of “intracellular homeostasis” on page 3 line 86
On page 6, 210 the authors refer to the study without citing or naming an author from that study. Please cite the relevant study the authors refer to.
General comments:
The English overall needs editing and improvements. Several abbreviations are not written full out or not correctly used. Example vascular endothelial growth factor is abbreviated as VEGR which is incorrect. Its VEGF and when talking about the receptor it’s abbreviated as VEGF receptor or VEGFR
Some words are capitalised while they are in the middle of a sentence, example is studies on page 3 line 108.
Some sentences are very long and needs to be rewritten to improve the readability. For example, page 4 lines 131 – 134 and page 7 lines 248 – 253
When discussing the current data, some parts are strongly focussed on mice, despite the authors discussing human data as well. This may be a personal preference, but when there is enough data available using human material. My personal opinion would be to focus on the human data as that is biologically more relevant.
Add sub-headings to improve readability on page 5, 7, 8, 9, 11 and 12. The fibroblast section is interesting but quite a long read without having proper headings. This would give a quick overview what the authors are going to discuss.
Overall, I would like to see more context. A lot of data is discussed and sometimes it feels it looks like a data dump. While the information is interesting and important, it does reduce the readability if the context is clearer and what does it mean, or how does it contribute to IPF.
The figures are good, however, I would like to see more figures if possible. Maybe need to split them in A-B-C and D for example. This would make it easier to put things in context and explain the complex signalling that is happening in IPF. As a final figure the authors could make a summarising graph to bring everything together.
Don’t use etc… in a sentence. Write several examples and leave it by that. Or If there is more to is, refer to other reviews that go deeper in on that specific topic.
Authors could be a bit more specific when they discuss extracellular matrix. Could specify a bit more which ECM proteins are involved as these proteins are important for cellular function in IPF.
Maybe the authors want to have a look at these 2 papers discussing ECM in context of IPF/fibrosis.
https://doi.org/10.1042/CS20190893 and https://doi.org/10.1111/febs.15282
page 10, line 392 the reference is missing. Authors speak about marrissa j et al but no reference is cited
A future perspective would be good to include, what should be the focus of research. Any interesting/promising pathways to look at.
Author Response
We wish to thank the reviewer for the thoughtful comments and helpful suggestions. We carefully considered all the comments and further revised and explained them to fully solve the problems raised by the reviewer in our revised version and point by point response.
Comment 1:Abstract (lines 32-33)
Need to remove the reference to COVID-19 as there is only 1 mentioning of COVID-19 in the manuscript itself and that one sentence is out of place as well. Reword the last sentence as well to better reflect the conclusion and/or take-home message for the readers.
Response 1: Thank you for your valuable comment. As suggested by the reviewer, the content of COVID-19 pandemic has been deleted, and the last sentence of the summary has been rewritten as required. Thank the reviewer for the suggestion.
Comment 2:Introduction:
First paragraph is written quite well with a good flow of information and why you write the review. The second and third paragraph is quite dense with information. This needs some rewording to improve the flow and improve readability for the reader.
Response 2: Thank you for your positive comment.We have applied for the editing service of your magazine and asked professional service staff to edit the manuscript. Thank you for your feedback.
Comment 3:On page 3 line 75 what do the authors mean with “work in their own right”?
Response 3: Thanks for the reviewer's question. I intended to express that different cells have their own functions and roles in the occurrence and development of IPF, but I did not express it well. I have asked professional service staff to help edit the paper, hoping to make it easier to understand.
Comment 4:Alveolar epithelial cells in pulmonary fibrosis:
What do the authors mean in the context of “intracellular homeostasis” on page 3 line 86
Response 4: Thanks for the reviewer's question. I intended to express “homeostasis of intracellular environment”. The metabolic activities of cells and the constant changes of the external environment will inevitably affect the nature of the internal environment, but the internal environment can maintain relative stability through the regulatory activities of the body, and the imbalance of the homeostasis of the internal environment can lead to diseases. I have asked professional service staff to help edit the paper, hoping to make it easier to understand.
Comment 5:On page 6, 210 the authors refer to the study without citing or naming an author from that study. Please cite the relevant study the authors refer to.
Response 5: Thanks for the reviewer's comment. In the revised version, reference 63 provides data support for P6, line226 (formerly P6, line210). Thank you for reviewing.
Comment 6:General comments:
The English overall needs editing and improvements. Several abbreviations are not written full out or not correctly used. Example vascular endothelial growth factor is abbreviated as VEGR which is incorrect. Its VEGF and when talking about the receptor it’s abbreviated as VEGF receptor or VEGFR
Response 6: I'm sorry for this mistake. I have corrected abbreviations and supplemented other abbreviations. Thank you for your reminding.
Comment 7:Some words are capitalised while they are in the middle of a sentence, example is studies on page 3 line 108.
Response 7: Thanks for the reviewer's reminding. AT1 of P3,line113 (original P3,line108) is capitalized as a normal acronym. I have checked the whole text to avoid such mistakes.
Comment 8:Some sentences are very long and needs to be rewritten to improve the readability. For example, page 4 lines 131 – 134 and page 7 lines 248 – 253
Response 8: Thank you for your comment.We have applied for the editing service of your magazine and asked professional service staff to edit the manuscript. I believe the English of the article will be improved and easier to understand.
Comment 9:When discussing the current data, some parts are strongly focussed on mice, despite the authors discussing human data as well. This may be a personal preference, but when there is enough data available using human material. My personal opinion would be to focus on the human data as that is biologically more relevant.
Response 9: Thanks for the reviewer's valuable advice. I have changed or added the inappropriate animal data in the article into human materials, and described the impact on IPF patients. Thanks for the reviewer's help.
Comment 10:Add sub-headings to improve readability on page 5, 7, 8, 9, 11 and 12. The fibroblast section is interesting but quite a long read without having proper headings. This would give a quick overview what the authors are going to discuss.
Response 10: Thank you for your positive comment. I have added subtitles to such complicated parts as epithelial cells, fibroblasts and immune cells. Thank the reviewer for the valuable suggestions.
Comment 11:Overall, I would like to see more context. A lot of data is discussed and sometimes it feels it looks like a data dump. While the information is interesting and important, it does reduce the readability if the context is clearer and what does it mean, or how does it contribute to IPF.
Response 11:Thanks for the reviewer's valuable advice. I am very sorry for the problems in my data discussion. The original intention of providing data is to provide support for background. I have made the data reference more reasonable and easy to understand by adding background or summative description.Thank you for your suggestion to make my article more complete.
Comment 12:The figures are good, however, I would like to see more figures if possible. Maybe need to split them in A-B-C and D for example. This would make it easier to put things in context and explain the complex signalling that is happening in IPF. As a final figure the authors could make a summarising graph to bring everything together.
Response 12: Thank you for your valuable comment. Figure 1 serves as a summary in the article. I have divided some complex diagrams into A-B-C small diagrams and explained the pathway in the diagram. Thank you for your help.
Comment 13:Don’t use etc… in a sentence. Write several examples and leave it by that. Or If there is more to is, refer to other reviews that go deeper in on that specific topic.
Response 13: I am very sorry for such irregularities. I have deleted the inappropriate etc., thanks for the reviewer's valuable advice.
Comment 14:Authors could be a bit more specific when they discuss extracellular matrix. Could specify a bit more which ECM proteins are involved as these proteins are important for cellular function in IPF.Maybe the authors want to have a look at these 2 papers discussing ECM in context of IPF/fibrosis.https://doi.org/10.1042/CS20190893 and https://doi.org/10.1111/febs.15282
Response 14: Thanks to the reviewer for the valuable comment, which enriched and complete the content of my paper. I also appreciate the reference provided by the reviewer, from which I learned a lot. I have added a specific description of the ECM protein in the hope that it will be helpful to this article.
Comment 15:page 10, line 392 the reference is missing. Authors speak about marrissa j et al but no reference is cited.
Response 15: Thanks for the reviewer's reminding. The reference of P11, line469 (original P10, line392) in the revised version is 139.
Comment 16:A future perspective would be good to include, what should be the focus of research. Any interesting/promising pathways to look at.
Response 16:Thank the reviewer for the helpful suggestion.The future focus is on how to bring hope for the treatment of IPF by regulating cell morphology and function and promoting fibrosis signaling pathways in vivo, combined with targeted cell and key pathway therapeutics.The promising pathways are the effect of TGF-β-related pathways on the intrinsic and extrinsic regulatory mechanisms of cells and the effect of intercellular communication on fibrosis, especially the effect of niche cells on the maintenance of alveolar stem cell function.Thank you for your review.
Round 2
Reviewer 1 Report
The manuscript has been improved and is sufficient for acceptance Advances in Respiratory Medicine.Reviewer 2 Report
Thank you very much for all the responses to my comments in the submitted review and any corrections made to the manuscript. I find the table summarising the contribution of different cell types to the pathogenesis of IPF very useful. Please make minor corrections for spelling errors and comma spacing etc. I believe that the manuscript can be published in its current version and is an interesting contribution to the discipline of IPF research.
Reviewer 3 Report
The authors did a great job in addressing my initial comments and I'm satisfied with all modifications by the authors. Three minor suggestions:
Page 15 Line 857 – it says IPF development and development. This need some correction
Page 16 line 883 – please explain what is crosslink stromal cells
Page 17 Lines 905 to 911 – please rewrite as both sentences start with "in the future". Would be good if it can be rewritten to small section.